# Comparative study of two Rift Valley fever virus field strains originating from Mauritania

**Mehdi Chabert[1,2,3], Sandra Lacôte[4], Philippe Marianneau[4], Marie-Pierre Confort[1], Noémie Aurine[5], Aurélie Pédarrieu[2,3], Baba Doumbia[6], Mohamed Ould Baba Ould Gueya[6], Habiboullah Habiboullah[6], Ahmed Bezeid El Mamy Beyatt[7], Modou Moustapha Lo[8], Jenna Nichols[9¤], Vattipally B. Sreenu[9], Ana da Silva Filipe[9], Marie-Anne Colle[10], Bertrand Pain[5], Catherine Cêtre-Sossah[2,3☯], Frédérick Arnaud[1☯], Maxime Ratinier[1☯] ***

**1** IVPC UMR754, INRAE, Universite Claude Bernard Lyon 1, EPHE, Université PSL, Lyon, France, **2** CIRAD, UMR ASTRE, Montpellier Cedex, France, **3** ASTRE, Univ Montpellier, CIRAD, INRAE, Montpellier, France, **4** ANSES, Virology Unit, Lyon, France, **5** Université Lyon 1, INSERM, INRAE, Stem Cell and Brain Research Institute, U1208, USC1361, Bron, France, **6** Direction des Services Vétérinaires, Ministère de l'élevage, Nouakchott, Mauritania, **7** ONARDEP, Nouakchott, Mauritania, **8** ISRA-LNERV, Dakar, Dakar Hann, Senegal, **9** MRC-University of Glasgow Centre for Virus Research, Glasgow, United Kingdom, **10** INRAE, Oniris, PAnTher, UMR703, Nantes, France

☯ These authors contributed equally to this work.
¤ Current address: School of Biodiversity, One Health and Veterinary Medicine, University of Glasgow, Glasgow, United Kingdom
* maxime.ratinier@univ-lyon1.fr

**Data Availability Statement:** All relevant data are within the manuscript and its Supporting Information files.

## Abstract

Rift Valley fever (RVF) is one of the major viral arthropod-borne diseases in Africa. In recent decades, RVF virus (RVFV), the causative agent of RVF, has been responsible for multiple outbreaks in West Africa with important consequences on human and animal health. In particular, an outbreak occurred in 2010 after heavy rains in the desertic region of Adrar, Mauritania. It was characterized by the appearance of severe clinical signs among dromedary camels. Another one occurred in 2013–2014 across Senegal and the southern part of Mauritania. In this study, we characterized two RVFV field strains isolated during these two outbreaks. The first strain, MRU25010-30, was isolated from a camel (2010) while the second, MRU2687-3, was isolated from a goat (2013). By deep-sequencing and rapid amplification of cDNA-ends by polymerase chain reaction, we successfully sequenced the complete genome of these two RVFV strains as well as the reference laboratory strain ZH548. Phylogenetic analysis showed that the two field viruses belong to two different RVFV genetic lineages. Moreover, we showed that MRU25010-30 replicates more efficiently in various *in vitro* cell culture models than MRU2687-3 and ZH548. *In vivo*, MRU25010-30 caused rapid death of BALB/c mice and proved to be more virulent than MRU2687-3, regardless of the route of inoculation (subcutaneous or intranasal). The virulence of MRU25010-30 is associated with a high viral load in the liver and serum of infected mice, while the death of mice infected with MRU2687-3 and ZH548 correlated with a high viral load in the brain. Altogether, the data presented in this study provide new avenues to unveil the molecular viral determinants that modulate RVFV virulence and replication capacity

**Funding:** This work was funded by a grant from Institut National de la Recherche pour l'agriculture, l'alimentation et l'environnement (INRAE, https://www.inrae.fr/en) GISA metaprogram (FORESEE project) awarded to CCS, FA and MR; and a World Organisation for Animal Health (WOAH, https://www.woah.org/en/home/) twinning project ISRA/CIRAD on RVF (2019-2023) awarded to AP, MML and CCS. MC Ph.D fellowship was co-financed by INRAE and Centre de coopération internationale en recherche agronomique pour le développement (CIRAD, https://www.cirad.fr/en). The funders did not play a role in the study design, data collection and analysis, decision to publish, or preparation of the manuscript.

**Competing interests:** The authors have declared that no competing interests exist.

## Author summary

Rift Valley fever is an arboviral zoonosis caused by Rift Valley fever virus (RVFV) belonging to the *Phlebovirus* genus. It poses a major risk for causing a public and animal health emergency and is a significant economic burden in many African countries. To date, our knowledge of the impact of RVFV genetic diversity on its virulence in mammalian hosts, replicative capacities, and transmission by mosquitoes is limited. In this study, we fully sequenced the genomes of two RVFV strains isolated in Mauritania during two distinct outbreaks (2010 and 2013) and showed that they were genetically distant. Interestingly, we showed that one of the strains (MRU25010-30) is able to replicate *in vitro* more efficiently than the other (MRU2687-3). Additionally, we showed that high levels of RNAemia and viral load in the liver are associated with rapid death in BALB/c mice infected with MRU25010-30, whereas mice infected by MRU2687-3 tend to die later with high viral load in the brain. In conclusion, our study confirms that RVFV strains from distinct genetic lineages have different phenotypic characteristics such as virulence and replication capacity. These data provide a strong basis for further studies aimed at identifying the viral genetic determinants responsible for the observed phenotypes.

## Introduction

Rift Valley fever virus (RVFV) is a zoonotic arbovirus transmitted by mosquitoes, mainly of the *Aedes* and *Culex* genera [1–3]. RVFV belongs to the *Phlebovirus* genus within the *Phenuiviridae* family of the *Bunyavirales* order and is endemic in several African countries [4]. Since its discovery in Kenya in 1930 [5], it caused major epidemics such as the ones in South Africa in 1951 [6], in Egypt in 1977 [7], and in Mauritania and Senegal in 1987 [8]. Rift Valley fever (RVF), the disease caused by RVFV, primarily affects domestic and wild ruminants such as goats, sheep, and cattle with high rates of morbidity and mortality [4]. Additionally, RVF is characterised by a high abortion rate in pregnant females and higher susceptibility of young animals compared to the adults [5,9]. Humans are usually infected by RVFV through either bites from infected mosquitoes or direct exposure to contaminated fluids during, for example, slaughter or care of sick animals [10,11]. In humans, RVFV infection generally causes an acute, self-limiting febrile illness. However, some patients may develop severe forms characterised by fulminant hepatitis associated with haemorrhages and, occasionally, retinitis or late-onset encephalitis [12–15]. Mice are particularly susceptible to RVFV infection. Therefore, they represent a good model to study RVFV pathogenesis as they develop symptoms comparable to severe human infections except for the haemorrhagic syndrome [12,16,17].

In Senegal, RVFV was first detected in 1987 and, since then, recurrent epidemics, interspersed with inter-epidemic periods, have occurred [8,18]. Mauritania has also been affected by several RVF outbreaks. Notably, in 2010, heavy rainfalls in the northern Adrar desertic region were followed by a RVFV outbreak with unexpectedly severe symptomatic forms and even death within the population of camelids that were considered as asymptomatic RVFV carriers [19,20]. Two types of severe forms have been described: (i) a hyperacute form leading within 24 hours to sudden death and (ii) an acute form causing fever, oedema on the neck, ocular discharge, blindness, nervous system disorder, and abortion as well as haemorrhagic symptoms leading to the death of the animal in few days. This RVFV epidemic also resulted in the death of 13 people among 63 human cases, although the number of cases is probably under reported [19]. Two years later, several southern regions of the country were again affected by

RVF, and the circulation of the virus lasted until 2015, concomitantly with a RVFV re-emergence in Senegal (2013–14) [21–23]. The last recorded outbreak occurred in 2020, emphasizing that RVFV strains are actively circulating in West Africa [24].

Until now, only one serotype has been described for RVFV. Although genetic diversity is considered low (between 1 and 5%), it is possible to separate RVFV strains into seven distinct genetic lineages (named A to G) [25]. Interestingly, it has been shown that distinct RVFV genetic lineages exhibit different virulence in the CD-1 mouse model [26]. RVFV genome is composed of three segments of single-stranded RNA with negative polarity. The large segment (L, 6404 nucleotides (nts)) encodes for the L protein, the viral RNA dependent RNA polymerase (RdRp). This protein is involved in the viral genome replication and transcription [27]. The medium segment (M, 3885 nts) allows the expression of several in-frame polyproteins that are subsequently matured by cellular proteases, such as signal peptidase, into functional viral proteins [28]. Depending on the nucleotides used as start codon, the RVFV Seg-M can be translated into four polyproteins encompassing: i) one long glycoprotein, named p78 or LGp (AUG1) and the glycoprotein Gc, ii) two non-structural proteins, NSm and NSm′ (AUG2/3) followed by the two surface glycoproteins Gn and Gc, and iii) only Gn and Gc (AUG4/5). The NSm proteins are located at the outer membrane of the mitochondria and display an antiapoptotic activity [29,30]. Moreover, mutant viruses unable to express NSm/NSm' are attenuated in Wistar-furth rats and C57BL/6J mice, indicating that NSm is a virulence factor of RVFV [31,32]. The two glycoproteins Gn and Gc are involved in the cell entry and morphogenesis of the viral particles [33]. The LGp/p78 is a glycoprotein known to be incorporated at the surface of insect cells-derived virions and it is essential for RVFV dissemination in mosquitoes [31]. Another study showed that the quantity of p78 produced by RVFV impacts its replication rates in human macrophages as well as its virulence in infected mice [34]. The small segment (S, 1690 nts) possesses two open reading frames (ORFs) and uses an ambisense coding strategy [35]. The first ORF located in the positive strand encodes for the nucleoprotein (N) that interacts with the viral RNA and contributes to the viral replication and transcription [27]. The second ORF, located in the negative strand, allows the expression of the non-structural protein S (NSs) described as the main virulence factor of RVFV [36–38].

Studies aiming to characterize circulating RVFV strains in West Africa are still very limited [39]. Here, we characterized, through *in silico*, *in vivo*, and *in vitro* approaches, two RVFV field strains isolated in Mauritania in 2010 (MRU25010-30, camel serum) and in 2013 (MRU2687-3, goat placenta). We fully sequenced the genomes of these two viral strains, determined their viral growth kinetics in several cell culture models, and assessed their virulence in BALB/c mice in comparison to the ZH548 reference strain. Our data showed that RVFV strains from distinct genetic lineages have strong phenotypic differences such as virulence in mice and replication capacity, and offers new perspectives to identify viral molecular determinants involved in RVFV replication and pathogenesis.

## Materials and methods

### Ethics statement

The experimental protocols complied with the regulation 2010/63/CE of the European Parliament and of the Council of 22 September 2010 on the protection of animals used for scientific purposes and as transposed into French law. These experiments were approved by the Anses/ENVA/UPEC ethics committee and the French Ministry of Research (Apafis n° 2018120710406718 (#17985)). All experiments with infectious RVFV have been conducted by trained staff in certified biosafety level 3 laboratories and by following strict biosafety protocols.

## Cell culture

A549, A549Npro (kindly provided by Richard E. Randall, University of St Andrews, UK), BSR (a clone of BHK21 cells, obtained from Karl K. Conzelmann, Ludwig-Maxmilians-Universität München, Germany) [40], and VeroE6 (purchased from ATCC) were cultivated in Dulbecco's modified Eagle Medium (DMEM; Gibco, Thermo Fisher Scientific, Villebon-sur-Yvette, France) supplemented with 10% heat-inactivated fetal bovine serum (FBS; GE HEALTHCARE Europe GmbH, Freiburg, Germany). HepaRG cells (human hepatic cell line purchased from Lonza, Colmar, France) were grown in William's E medium (Gibco) supplemented with 10% FBS, 1% L-glutamine (Gibco), 0.5 μM hydrocortisone (Sigma-Aldrich, Merck, Saint-Quentin Fallavier, France), and 5 μg/mL insulin (Gibco).

Two independent isolates of human-induced pluripotent stem cells (hiPSCs), declared onto the CODECOH platform with the number DC-2021-4404, were maintained in mTeSR1 (StemCell Technologies, Saint Egreve, France) on a matrigel coating (Corning, Thermo-Fisher, Illkirch-Graffenstaden, France), and dissociated with dispase (StemCell Technologies), according to the manufacturer's instructions. These cells were subsequently differentiated into neurons, astrocytes, and oligodendrocytes for 45 days as previously described [41]. Two independent differentiation experiments were assessed for each isolate.

All cell lines were grown in humidified atmosphere of 5% $CO_2$ at 37˚C.

## Virus culture

The MRU25010-30 strain was isolated from the serum of a sick camel sampled in Lemsayddi, within the desertic Adrar region of Mauritania during the 2010 outbreak [19,42], at "Institut Sénégalais de Recherches Agricoles—Laboratoire National de l'Elevage et de Recherches Vétérinaires" (Senegal). The MRU2687-3 strain was isolated, at the "French Agricultural Research Centre for International Development" (France), from the placenta of an aborted goat sampled in the Brakna region of Mauritania along the Senegal river basin, during the outbreak of 2013 [22]. These two strains were passaged twice in VeroE6 before being used in this study. The ZH548 strain was isolated in Egypt, from serum of a febrile human patient during the 1977–1978 outbreak and was used in this study as a reference strain. The cell culture passages history of ZH548 strain used in this study is undetermined.

All RVFV strains stocks were produced by infecting VeroE6 cells grown in DMEM supplemented with 4% FBS. Two- or 3-days post-infection (pi), the supernatants were harvested, clarified by centrifugation at 500g for 5 minutes (4˚C), aliquoted and stored at -80˚C. The titrations were performed by plaque assays using VeroE6 cells as previously described [41].

## Rapid amplification of cDNA-ends by polymerase chain reaction (RACE-PCR)

Viral RNA of MRU25010-30, MRU2687-3, and ZH548 viral stocks were extracted using the QIAmp Viral RNA kit (Qiagen; Courtaboeuf, France) according to the manufacturer's instructions. PolyA tails were added to the viral genomic or antigenomic RNA molecules using the mMESSAGE mMACHINE T7 Transcription Kit (Ambion, France) according to the manufacturer's instructions. The reverse transcription was performed using the PrimeScript RT reagent kit (Takara, Saint-Germain-en-Laye, France) with a primer oligo-d(T)-AP (5' GAC CACGCGTATCGATGTCGACTTTTTTTTTTTTTTTTTTv 3'). PCR DNA amplifications were performed with CloneAmp HiFi PCR premix (Takara) using Primer-AP (5' GACCACGCGT ATCGATGTCGAC 3') and specific primer for each end (5' UTR S: 5' TAGTCCCAGTGACA GGAAGC 3'; 3' UTR S: 5' GGTATCCTGGGAGGACCAT 3'; 5' UTR M: 5' GCCATGGTTTC

TCTCCCTAT 3'; 3' UTR: 5' TAGGCGGGAAGCAGGGGG 3'; 5' UTR L: 5' CTGGAATGCA
CCTCTTTCATCTC 3'; 3' UTR L: 5' TGGATGTTAGTGGCCCTTACG 3'). The PCR ampli-
cons were gel purified with NucleoSpin gel and PCR clean-up kits (Macherey-Nagel, Ger-
many) following the manufacturer's protocol, and subsequently analysed by Sanger
sequencing (Eurofins Genomics, Germany).

## Deep-sequencing analysis

RNA from RVFV stocks used to infect mice (see below) were extracted using TRIzol Reagent
(Invitrogen), further purified using Direct-zol RNA MiniPrep kit (Zymo Research, Irvine,
USA), according to the manufacturer's protocol. cDNA was synthesised using SuperScript III
(Thermo Fischer Scientific, Part Number 18080044) and NEBNext Ultra II Non-Directional
RNA Second Strand Synthesis Module (New England Biolabs, Part Number E6111L) as per the
manufacturer's instructions. For library preparation, the Kapa LTP Library Preparation Kit for
Illumina Platforms was used (Kapa Biosystems, Part Number KK8232). The cDNA was pro-
cessed through the End Repair step and the subsequent library preparation protocol according
to the manufacturer's guidelines until the Adapter Ligation step. Briefly, following bead-based
clean-up, the cDNA underwent End Repair at 20°C for 30 minutes. This was followed by
another bead-based clean-up and A-tailing at 30°C for 60 minutes. After a third bead-based
clean-up, the DNA fragments were incubated with NEB's diluted adaptor (0.15 μM) to accom-
modate the low nucleic acid input at 20°C for 60 minutes, followed by treatment with USER
enzyme at 37°C for 15 minutes. At this stage the samples were uniquely indexed using the NEB-
Next Multiplex Oligos for Illumina 96 Unique Dual Index Primer Pairs Sets 1 and 2. 16 cycles
of PCR were performed for all samples. Libraries were quantified by Qubit (Thermo Fisher Sci-
entific), with size profile determined by TapeStation (Agilent, Santa Clara, USA). Libraries were
pooled at equimolar concentrations and sequenced on an Illumina Microcartridge, 300 cycles.

## Bioinformatic analysis

There were a total of 2067461, 1598621, and 905582 reads (S1 Table) obtained from high-
throughput sequencing from samples MRU25010-30, MRU2687-3, and ZH548, respectively.
Quality filtering of the reads was performed using Trim Galore (https://github.com/
FelixKrueger/TrimGalore) program with minimum PHRED score 25 and read length 75 nts.
Filtered reads were mapped to the reference sequences (NC_014395, NC_014396, and
NC_014397) using Tanoti (https://github.com/vbsreenu/Tanoti) and consensus sequences
were generated using Sam2Consensus (https://github.com/vbsreenu/Sam2Consensus) pro-
grams. All the reads with a minimum Phred score of 25 were used in the consensus calling.
The minimum read depth used for consensus generation is 20 reads. In the genome terminal
sequences, consensus sequences were curated with the majority read consensus. The low-fre-
quency variants from the SAM (sequence alignment map) file were calculated using the Sam-
eer program (https://github.com/vbsreenu/Sameer). All the amino acid variants with a read
depth greater than 20 reads and frequency more than 5% are reported in our study. The assem-
bled sequences for L, M, and S segments, combined with RACE-PCR data, were deposited in
GenBank with accession numbers OR844390, OR844391, OR844392 (MRU25010-30),
OR844387, OR844388, OR844389 (MRU2687-3), and OR805805, OR805806, OR805807
(ZH548).

## Phylogenetic and protein analysis

Nucleotide and protein sequences were aligned using ClustalW program [43] or codon
method in MEGA X software [44]. The evolutionary history was inferred using the maximum

likelihood (ML) method and general time reversible model (GTR) [45] implemented in the above-mentioned MEGA X software. To compare the genetic relatedness of the sequenced viruses, phylogenetic analyses were performed against a panel of 33 ancestral complete nucleotide sequences described by Bird and colleagues [25] and SPU77/04 strain sequences (Namibia, 2004; accession numbers: KY196500 (S), KY126703 (M), and KY126678 (L)) [46].

### *In vitro* viral infection

HepaRG, A549 and A549Npro were seeded in 12-well plates and infected at a multiplicity of infection (MOI) of 0.01 in 4% FBS DMEM medium. Two hours pi the inoculum was removed and replaced by fresh medium. The supernatant was harvested at 24h and 48h pi. Differentiated hiPSC cells were infected with $2\times10^4$ plaque-forming units (PFU, MOI of approximately 0.1) in 4% FBS DMEM medium. The inocula were washed 2h pi and hiPSC maturation medium was added. The supernatant was harvested at 24 and 48h pi. Viral titres were measured in all supernatants by endpoint dilution method using BSR cells and expressed as 50% tissue culture infectious dose ($TCID_{50}$/ml) determined by Spearman-Kärber method. Each experiment was conducted independently two times (neural differentiated hiPSC cells, each time with cells from two different donors) or three times (in duplicate or triplicate: HepaRG, A549, and A549Npro) using at least two different stocks of each virus. Statistical analyses were conducted at each time point using Kruskal–Wallis test and Graphpad Prism 8.4 (Graphpad Software Inc., La Jolla, CA, USA).

### *In vivo* studies

Six- to 8-weeks old female BALB/c mice (purchased from Janvier Labs; Le Genest St Isle, France) were infected either intranasally (IN) or subcutaneously (SC) as previously described [41]. RVFV doses used in this study were either $10^1$ or $10^3$ PFU. Volumes of infections were 20 µl per nostril for IN inoculation and 100 µl for SC inoculation. Mock group contained two or three animals, each of the RVFV-infected group contains six animals (MRU25010-30, MRU2687-3, and ZH548) and experiment was repeated twice except for ZH548. Weight and temperature of mice were monitored every day. The sera of infected mice were collected at day 3, 6, 10, and 15 pi for RT-qPCR analysis (see below). Mice showing severe clinical signs of RVFV were ethically euthanized under anaesthesia. All surviving mice were euthanized at the end of the experiment. Brain and liver of euthanized mice were collected and used to perform RT-qPCR. Viral RNA-positive brain and liver samples were also titrated by plaque assays, and viral titres were expressed as PFU/g. Statistical analyses (Gehan-Breslow-Wilcoxon test) were performed by using the Graphpad Prism software (Graphpad Software Inc).

Additionally, 6 to 8 weeks old female BALB/c mice were IN or SC infected with $10^3$ PFU (MRU25010-30, MRU2687-3, and ZH548). Mice were euthanized at D1, D2, and D3 pi (MRU25010-30), at D3 and D6 pi (MRU2687-3), and at D2, D3, and D6 pi (ZH548). Serum, liver, and brain of euthanised mice were collected and analysed by RT-qPCR. Each group contained three animals.

### RT-qPCR

Brain and liver of euthanized mice were weighed and homogenized in 500 µl of DMEM with two stainless steel beads (Thermo Fisher Scientific) three times for 30 s at 30 Hz using Tissue Lyser II (Qiagen). After centrifugation at 2,000 rpm for 5 min, supernatants were collected. Viral RNA purification from serum, liver, and brain samples was performed using QIAmp Viral RNA (Qiagen) according to the manufacturer's instructions. One µg of purified RNA was treated twice with 5 µl of RNAse-free DNAse (Qiagen) and subjected to a final on column

purification. Part of the RNA (5μL) was reverse transcribed into cDNA using the SuperScriptIII Platinum One-Step Quantitative RT-PCR System (Thermo Fisher Scientific). The cDNA (3 μL, duplicates) was subjected to a PCR reaction targeting the M segment by using the primer RVs 5'—AAA GGA ACA ATG GAC TCT GGT CA -3', primer RVAs 5'—CAC TTC TTA CTA CCA TGT CCT CCA AT -3' and a RVP probe 6-FAM AAA GCT TTG ATA TCT CTC AGT GCC CCA A -TAMRA [47]. The PCR cycling conditions were 95˚C for 5 min, followed by 45 cycles set up as follow: 95˚C for 5 s and 57˚C for 35 s. The number of viral RNA copies in each sample was determined using Gn RNA standard calibration curve, as previously described [48]. The results were expressed as number of viral RNA copy per ml of serum or g of tissue (liver or brain).

The numerical data used in all figures are included in S1 Data.

## Results

### RVFV field isolates clustered in two distinct genetic lineages

Whole genome sequences of MRU25010-30, MRU2687-3, and ZH548 strains were obtained by deep sequencing (S1 Table and S1–S3 Figs). The sequences of the 5' and 3' ends of each segment were confirmed by RACE-PCR analysis. We found that the percentage of identity between the nucleotide sequences of the two Mauritanian field strains was 94.6% for the L segment (6404 nts), 94.6% for the M segment (3885 nts), and 95.7% for the S segment (1691 and 1690 nts). Notably, the length of the intergenic region within the S segment of MRU25010-30 was 1 nt longer than MRU2687-3 (and ZH548). Using the phylogenetic analysis proposed by Bird and colleagues [25], MRU25010-30 clustered with sequences of lineage E whereas MRU2687-3 was closely related to lineage A, the latter including ZH548 strain (Fig 1A). It is important to note that, for both strains, the sequences of the three segments clustered in the same lineage, indicating that these strains are not the result of a reassortment event between two viruses from different lineages. At the amino acid level, the percentage of identity between MRU25010-30 and MRU2687-3 strains was 99.0% for L protein, 99.1% for the full-length M polyprotein (starting from AUG1), 100% for the N, and 98.5% for the NSs. All amino acid residue substitutions between MRU25010-30 and MRU2687-3 are represented in the Fig 1B and listed in S2 Table which also includes those of the ZH548 strain. Next, we focused on the amino acids differing between MRU25010-30 and the other two strains MRU2687-3 and ZH548. By classifying amino acid residues based on the polarity of their R-group (non-polar, polar with no charge, polar with negative charge, and polar with positive charge), we identified six amino acid residue substitutions of potential interest: R42G (NSm), K384T (Gn), D157G, D407G, G411S, and T2033A (L).

Deep-sequencing approach also provided us with the intra-strain genetic diversity. Indeed, nucleotide diversity can be measured using the Shannon entropy calculation for each position (S1–S3 Figs). We were able to identify sites as polymorphic and focused on the ones leading to an amino acid residue substitution (Table 1). We observed low intra-strain genetic diversity for N, NSs, Gc, and L proteins, whereas Gn protein sequence was more impacted than the other proteins. For instance, the residue at position 431 (asparagine, N) existed as a population for both MR2687-3 and ZH548 with either a lysine (8%) or a serine (9%), respectively. Interestingly, the consensus sequence of ZH548 strain had two leucine (L) at position 232 (Gn, 83%) and 747 (Gc, 88%). This sequence coexisted with minor subpopulations harboring a glutamine (232Q) and an isoleucine (747I), these last two amino acid residues being conserved for the two field strains. Notably, the amino acid residue at position 384 of MRU25010-30 was highly variable and could be either a lysine (K, 49%), a threonine (T, 43%, conserved in the MRU2687-3 and ZH548 strains) or an arginine (R, 7%). Finally, data obtained from both

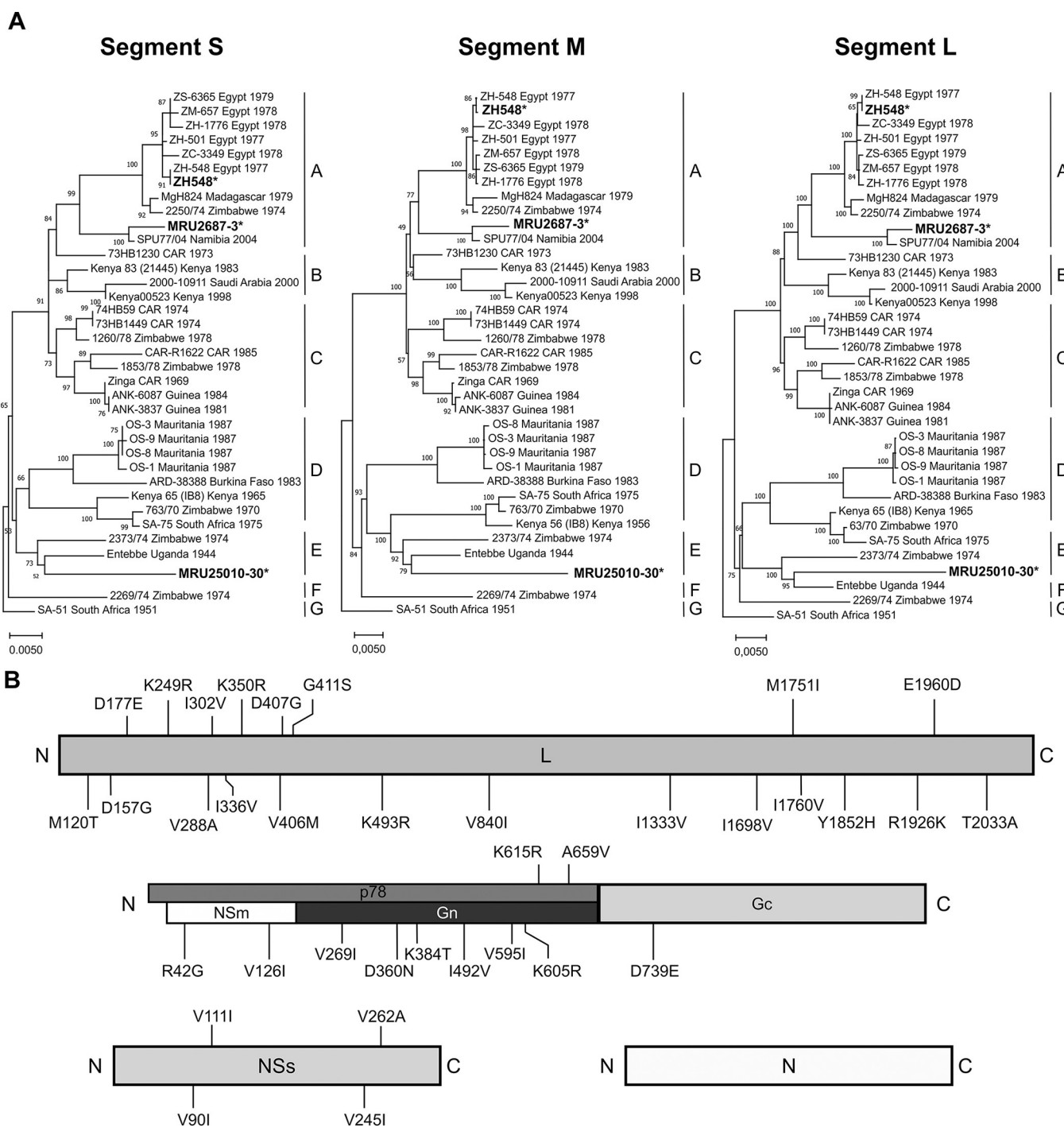

**Fig 1. Phylogenetic trees and protein sequences analysis of MRU25010-30 and MRU2687-3 strains.** (A) Phylogenetic analysis of the S, M, and L segments based on the classification of Bird and colleagues [25]. The trees were generated by the Maximum likelihood GTR model with 500 bootstraps. MRU25010-30, MRU2687-3, and ZH548 are in bold text and indicated by an asterisk. Genetic lineages (A-G) originally proposed by Bird and colleagues are represented for each segment. (B) Schematic representation of the amino acid residue substitutions between the consensus sequences of MRU25010-30 and MRU2687-3. L protein (top panel), full length polyprotein of segment M (middle panel) and, NSs and N proteins of segment S (bottom panel) are presented. p78 is shown in dark grey, NSm in white, Gn in black, and Gc in light grey. Position and amino acid residues of MRU25010-30 (left) and MRU2687-3 (right) strains are indicated.

**Table 1. Intra-strain genetic diversity of MRU25010-30, MRU2687-3, and ZH548 strains.**

| Strain | Segment | Protein | AA Position | Consensus AA (relative frequency, %) | Others AA (relative frequency, %) |
|---|---|---|---|---|---|
| MRU25010-30 | S | N | 216 | E (85) | G (15) |
| | M | Gn | 384 | K (49) | T (43) R (7) |
| MRU2687-3 | S | NSs | 112 | S (90) | P (10) |
| | M | Gn | 376 | E (82) | G (18) |
| | M | Gn | 430 | A (79) | T (21) |
| | M | Gn | 431 | N (90) | K (8) |
| | M | Gn | 642 | N (92) | S (8) |
| | L | L | 301 | A (92) | E (7) |
| ZH548 | M | Gn | 232 | L (83) | Q (17) |
| | M | Gn | 236 | G (93) | R (7) |
| | M | Gn | 431 | N (91) | S (9) |
| | M | Gc | 747 | L (88) | I (12) |

Amino acid residue substitutions are classified by strains, segments, and related proteins. Note that the numbering of Gn and Gc proteins starts from AUG1 used to translate p78. The relative frequencies, expressed in percentage, were calculated using read counts at the given position and rounded to the nearest unit. AA, amino acid; S, small; M, medium; L, large.

RNA-sequencing and RACE-PCR approaches allowed us to identify genetic diversity within the 5'UTR of the anti-genomic M segment for both MRU2687-3 and ZH548 strains (S4 Fig). Indeed, the nucleotide at the position 10 was either an uracil (U, preponderant in MRU2687-3 for both RNA-sequencing and RACE PCR data) or a cytosine (C, preponderant in ZH548 for RNA-sequencing but under-represented in RACE-PCR, S4 Fig), the latter being constituently found in RNA-sequencing for MRU25010-30 strain.

## MRU25010-30 replicates more efficiently than MRU2687-3 in various *in vitro* cell culture models

We next assessed the replication kinetics of these three RVFV strains in different human cell models, including some from organs targeted by RVFV (liver and brain). We observed that human hepatic cells (HepaRG) infected with MRU25010-30 produced significantly more infectious viral particles over time than those infected with MRU2687-3, with a difference in viral titres of approximately 1.5 and 0.5 $\log_{10}$ ($TCID_{50}$/ml) at 24h and 48h pi, respectively (Fig 2). Notably, ZH548 viral titre was comparable to MRU25010-30 at 24h pi but the latter became 5.6 times lower at 48h pi, like that of MRU2687-3. These results suggested that MRU25010-30 replicates more efficiently in hepatic cells than the two other strains. Next, we used human induced pluripotent cells (hiPSC) differentiated into "neural cells", essentially composed of neurons, astrocytes, and oligodendrocytes [41]. At 24h pi, MRU25010-30 and ZH548 displayed higher infectious titres than MRU2687-3 (Fig 2). However, at 48h p.i., MRU25010-30 produced average titres approximately 2.4 $\log_{10}$ ($TCID_{50}$/ml) higher than those observed for MRU2687-3 and ZH548 (Fig 2), indicative of its enhanced ability to replicate in these cells.

In mammals, the interferon system is often considered as the first line of defence against viral infection [49]. To assess the impact of the interferon response on the *in vitro* growth properties of the RVFV strains studied, we infected A549 (known to produce and respond to type 1 interferon, IFN-I) and A549Npro (expressing Bovine viral diarrhoea virus N-terminal protease that is blocking IRF3 activity and therefore inhibiting IFN-I production) cells [50]. We next measured the viral titres at 24h and 48h pi. Notably, we showed that all RVFV strains

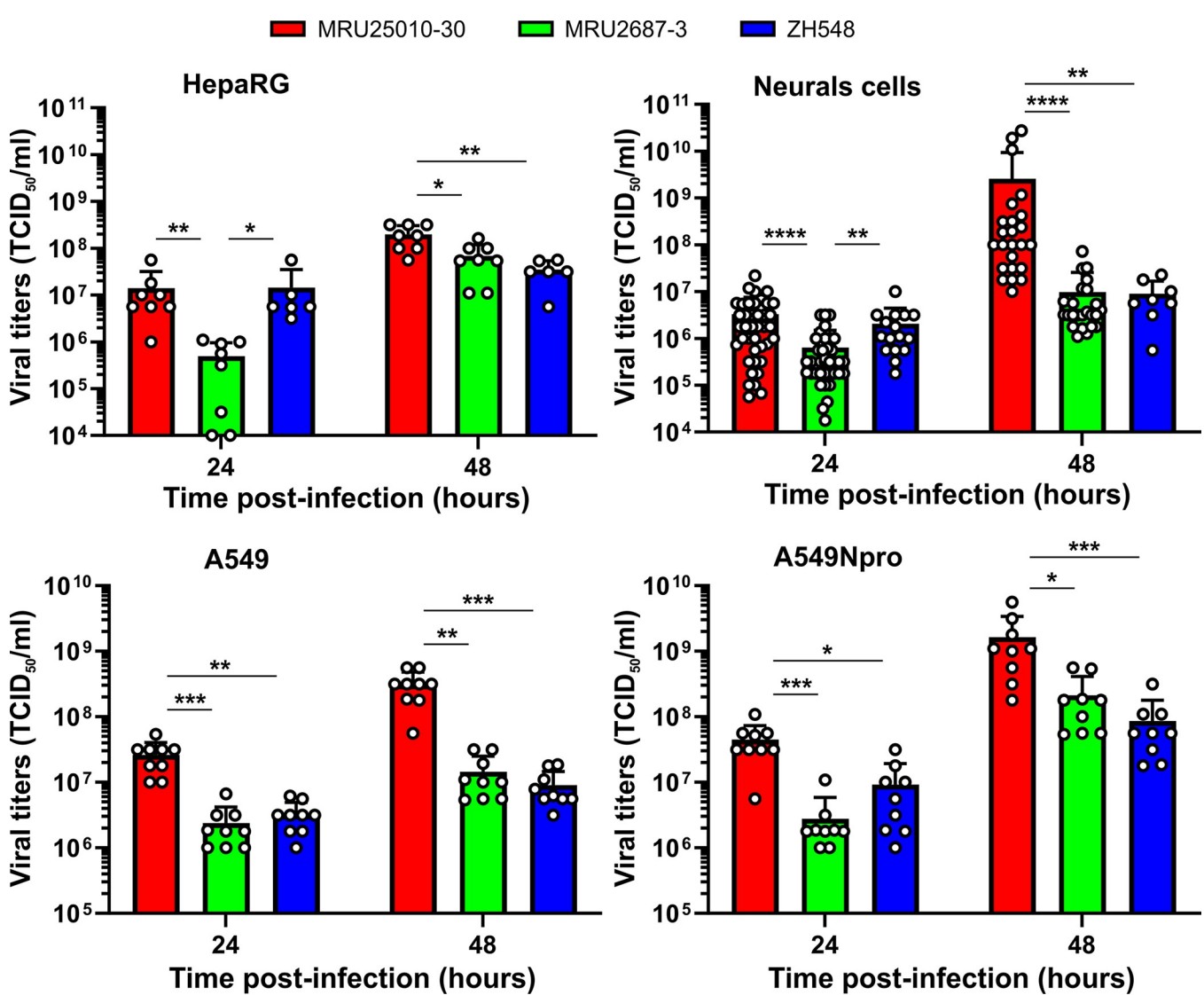

**Fig 2.** *In vitro* **growth properties of MRU25010-30, MRU2687-3, and ZH548 strains.** HepaRG, A549, and A549Npro cells were infected at MOI 0.01, while hiPSC differentiated into neural cells were infected with $2x10^4$ PFU (approximately a MOI of 0.1). Supernatants were analysed by $TCID_{50}$ method at indicated time pi and titres are expressed as $TCID_{50}$/ml of supernatant. In conventional cell lines, this experiment was repeated three times independently, each time either in duplicate or in triplicate. Experiment in hiPSC differentiated into neural cells was repeated twice independently (MRU25010-30 and MRU2687-3) or performed once (ZH548) using hiPSC obtained from two donors (MRU25010-30 and MRU2687-3: n = 40 and n = 38 at 24h pi, respectively, and n = 24 at 48h pi; ZH548: n = 16 at 24h pi, and n = 8 at 48h pi, n corresponding to the number of supernatant analysed). Error bars represent standard deviations around the mean value. Kruskal-Wallis statistical analyses were performed at each time point, and statistical significance is presented as follows: p < 0.05 (*), p < 0.01 (**), p < 0.001 (***), and p < 0.0001 (****).

produced, on average, between 5 to 15 times more infectious viral particles at 48h pi in A549Npro cells compared to A549 cells, indicating that the IFN response partially inhibits RVFV replication (Fig 2). However, MRU25010-30 strain consistently reached significantly higher titres than MRU2687-3 and ZH548 strains in the two cell lines (Fig 2), suggesting that the interaction of these viruses with host IFN-I response is not the key factor responsible for the difference of viral fitness observed between MRU25010-30 and the two other strains, MRU2687-3 and ZH548.

## RVFV MRU25010-30 is more virulent than MRU2687-3 in BALB/c mice

We next experimentally infected BALB/c female mice with MRU25010-30 and MRU2687-3 strains as well as the reference strain ZH548, in order to investigate their virulence in immunocompetent mice. Viruses were inoculated by two different routes: either subcutaneously (SC) or intranasally (IN). We used two different infectious doses for SC groups ($10^1$ and $10^3$ PFU) and only the highest dose for IN groups ($10^3$ PFU).

The Kaplan-Meier curves showed that MRU25010-30 strain was significantly more virulent than MRU2687-3 in the three conditions tested (SC-$10^1$, $p<0.001$; SC-$10^3$, $p<0.001$; IN-$10^3$, $p<0.01$) (Fig 3). Remarkably, the onset of mortality of MRU25010-30 infected mice occurred much earlier (D3/4 pi) compared to those infected with MRU2687-3 (approximately D8 pi). Only one mouse infected with MRU2687-3 (SC-$10^3$ group) died prematurely at D5 pi. Overall, the median survival of mice infected with MRU25010-30 was 4–5 days while those infected with MRU-2687 was 10 days. Although ZH548 (n = 6) induced higher mortality rates than MRU2687-3, both behaved similarly when inoculated subcutaneously, with a median survival of 9 days for mice infected by ZH548, regardless of the initial dose (SC-$10^1$ and SC-$10^3$). Unlike the other two strains, ZH548 was more virulent by intranasal inoculation than by subcutaneous injection (p = 0.0411; median survival = 8 days) and was statistically similar to MRU25010-30 in this condition. Additionally, most probably due to the sudden death induced by MRU25010-30 strain (SC-$10^1$ and SC-$10^3$ groups), we did not observe an impact of the infection on body weight (S5 Fig). In contrast, after infection with MRU2687-3 or ZH548, we observed a reduction in mice body weight prior to death of the animals similar to mice infected intranasally with MRU2010-30 (S5 Fig). Overall, our data showed that MRU25010-30 strain is more virulent than MRU2687-3 and is causing sudden death of BALB/c mice.

## Early death induced by MRU25010-30 strain is associated with high RNAemia

We then determined the level of viral RNA in the sera of the mice over time by RT-qPCR (Fig 4). Whatever the inoculation conditions, we observed that mice infected with MRU25010-30 had, on average, the highest number of genome copies per ml of serum at D3, with values up to $10^{12-13}$ genome copies per ml of serum for the SC-$10^3$ group. In comparison, these values were 1 to 2 $\log_{10}$ lower for mice infected with MRU2687-3, with peak values between $10^{11-12}$ genome copies per ml of serum. Concomitantly, a lower dose or an intranasal infection tended to delay the RNAemia peak and the onset of death, as shown in the Fig 3. Overall, the RNAemia induced by ZH548 were comparable to that of MRU2687-3 strain, although slightly lower at D3 pi. We did not observe marked differences in ZH548 viral loads between SC-$10^3$ and IN-$10^3$ groups, even if the route of infection modulated the virulence of this strain (Fig 3). Interestingly, although high viral RNA copy numbers were observed, at early time points, in the sera of mice infected by MRU2687-3, the vast majority of the animals showed a reduced or no RNAemia in the sera at the time of death.

In parallel, we also investigated the dynamics of seroconversion in RVFV infected mice (S6 Fig). We did not observe striking differences between all the groups as most of the mice produced anti-RVFV IgM by the end of the experiment (if not killed by the infection), except for one and five mice infected by MRU2687-3 from SC-$10^3$ and SC-$10^1$ groups, respectively. In SC-$10^1$ group, one of the five mice that failed to produce anti-RVFV IgM showed a transient RNAemia at D3 pi, whereas the other four mice failed to produce detectable RNAemia in their sera, suggesting that MRU2687-3 inoculation did not lead to a productive RVFV infection in these animals. Overall, these data showed that early death induced by MRU25010-30

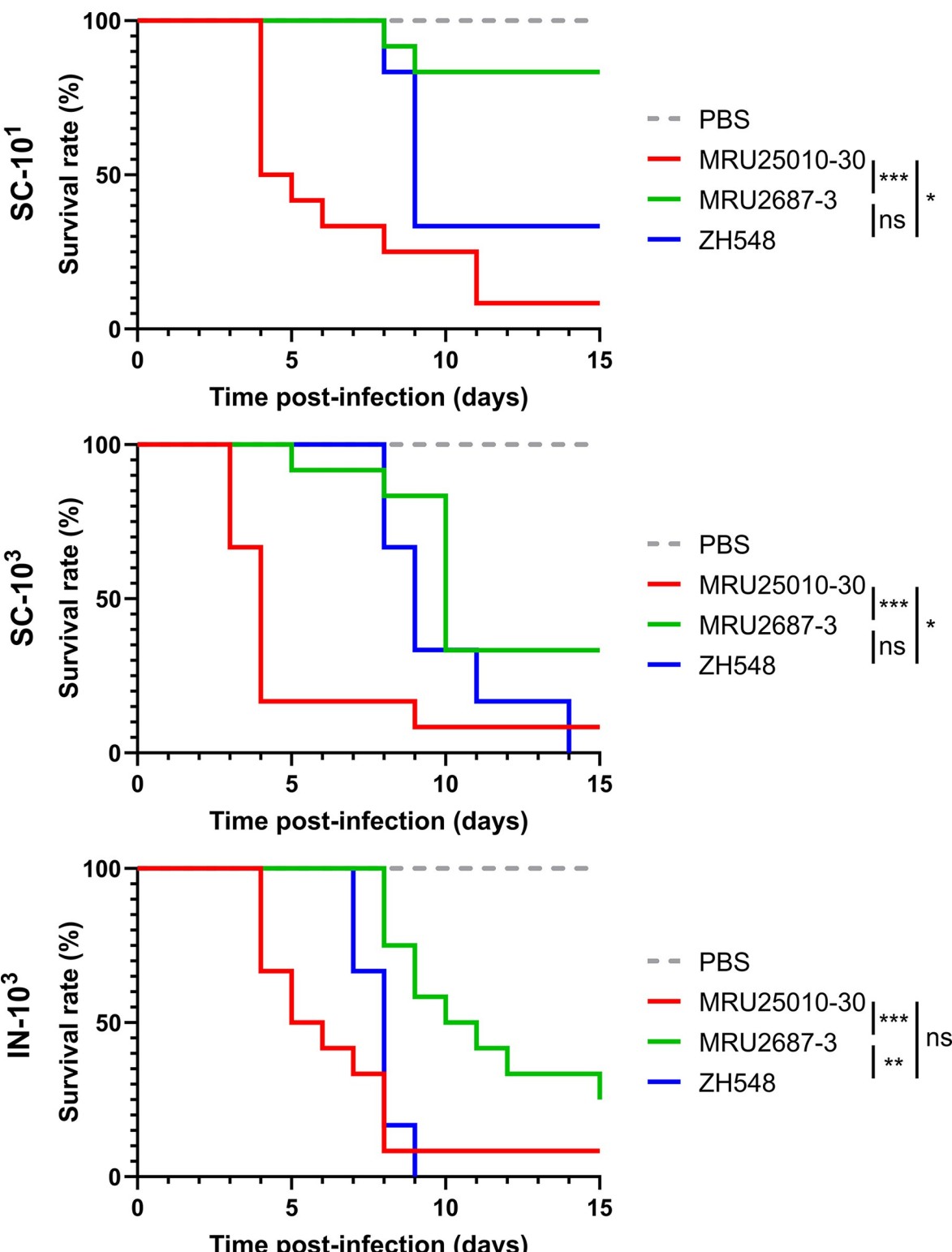

**Fig 3. Kaplan-Meier survival curves of BALB/c mice infected by MRU25010-30, MRU2687-3, and ZH548 strains.** 6–8 weeks old female BALB/c mice were infected subcutaneously (SC) or intranasally (IN) with $10^1$ or $10^3$ PFU of the indicated RVFV strains (n = 12 for MRU25010-30 and MRU2687-3 strains, n = 6 for ZH548 and n = 5 for PBS). Survival curves were analysed using the Gehan–Breslow–Wilcoxon test (ns p>0.05; * p<0.05; ** p<0.01; *** p<0.001).

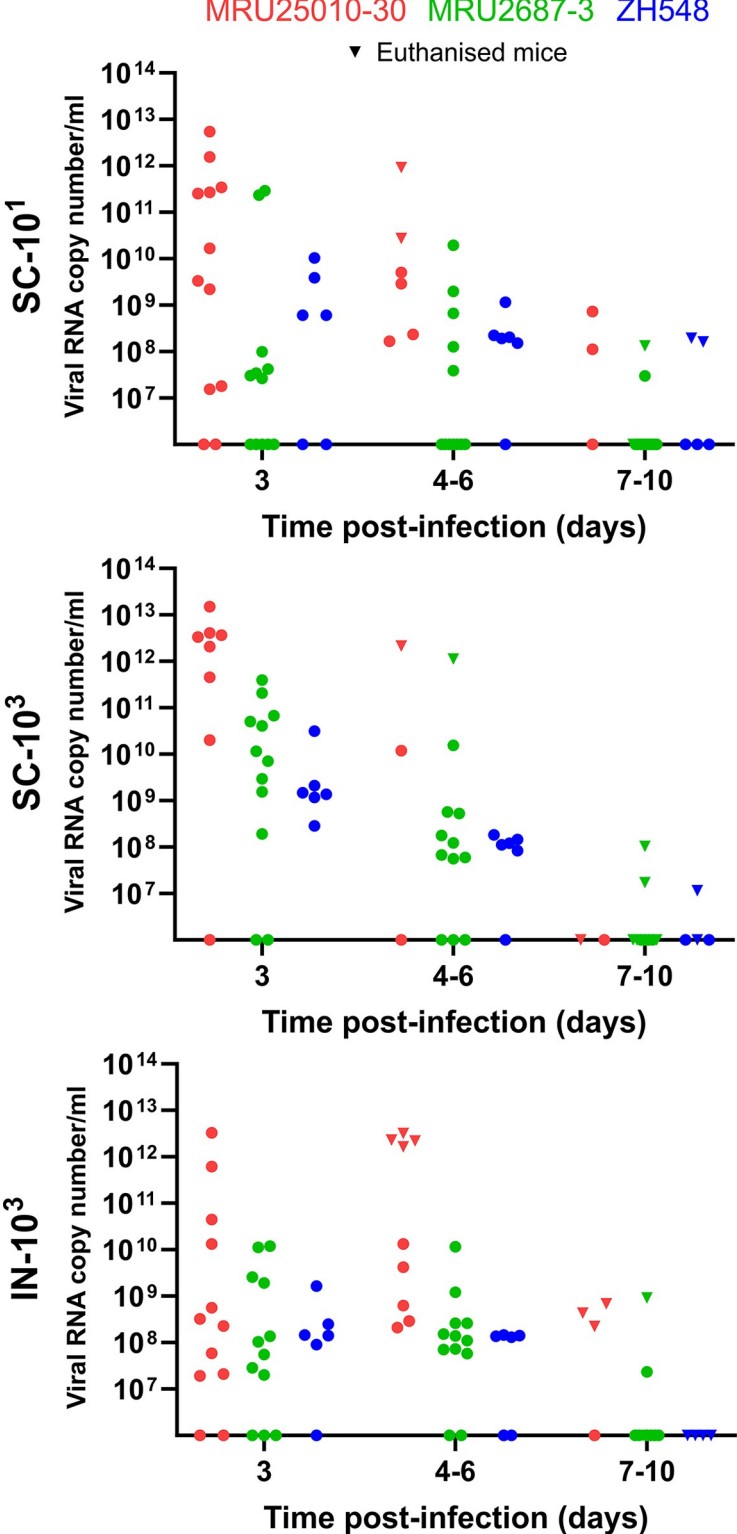

**Fig 4. RVFV RNAemia kinetics in BALB/c mice.** 6–8 weeks old female BALB/c mice were infected intranasally (IN) or subcutaneously (SC) with $10^1$ or $10^3$ PFU of the indicated RVFV strain. Levels of viral RNA were measured in sera of infected mice by RT-qPCR targeting segment M. Values are expressed in viral RNA copy number per ml of serum. Sera were collected at days 3, 6, and 10 (circles) as well as in euthanised mice (inverted triangles). MRU25010-30 (n = 12) is indicated in red, MRU2687-3 (n = 12) in green, and ZH548 (n = 6) in blue.

correlated with high viral load in mice sera, whereas mice dying from MRU2687-3 infection at later time points displayed reduced RNAemia.

## RVFV viral load in the liver and brain of BALB/c infected mice

In order to further characterise RVFV dissemination into organs, we assessed the level of RVFV RNAs in the liver and brain of the mice that had been euthanized during the course of the experiment (Fig 5). Because of the sudden death of the majority of mice in the MRU25010-30 SC-$10^3$ group, only one animal could be analysed at D4 pi. In this mouse, the virus was detected both in the liver ($\approx 10^{11}$ vRNA copies/g) and, to a lesser extent, in the brain ($\approx 10^9$ vRNA copies/g). The same observation was made in mice from: (i) SC-$10^1$ group ($\approx 10^{10-12}$ versus $\approx 10^9$ copies of vRNA/g, D3-6 pi), (ii) IN-$10^3$ group ($\approx 10^{11-12}$ versus $\approx 10^{8-9}$ copies of vRNA/g, D3-6 pi) (Fig 5). In contrast, at later time points (D7-D15 pi, squares and dots in Fig 5), although MRU25010-30 RNA was detected in both organs, the viral RNA copy numbers were higher in the brain than in the liver, similarly to what we observed with MRU2687-3 and ZH548 strains regardless of the inoculation route (Fig 5). Next, we tested the presence of infectious RVFV particles in the viral RNA-positive liver and brain samples by plaque assays. Overall, we observed similar patterns between viral infectious titres and RT-qPCR data (Fig 5). Altogether, these data suggested that at the early stages of infection (D3-6 pi), the virulence of MRU25010-30 is mainly related to a high viral load in the liver, even though the virus was also detected very early in the brain of some of the infected animals. In contrast, the pathogenicity of MRU2687-3 and ZH548 mainly correlated with a high viral load in the brain of infected BALB/c mice.

Next, we refined the viral replication kinetics and organ tropism at early time points after inoculation of the three RVFV strains. To do so, we infected BALB/c mice intranasally (IN-$10^3$) or subcutaneously (SC-$10^3$) and collected serum, liver, and brain at D1, D2, and D3 pi (MRU25010-30), at D3 and D6 pi (MRU2687-3), and at D2, D3, and D6 pi (ZH548). After SC infection, MRU25010-30 was detected in the sera and the liver of two out of three mice at D2 ($10^{8-9}$ vRNA copies/ml and $10^{8-9}$ vRNA copies/g) and all the mice at D3 ($\approx 10^{9-12}$ vRNA copies/ml and between $10^{10-13}$ vRNA copies/g) (Fig 6). Interestingly, MRU25010-30 was also found in the brain of one mouse at D1 pi ($\approx 10^8$ vRNA copies/g) and two mice at both D2 and D3 pi. These data confirmed that MRU25010-30 is replicating very early after infection in both the liver and the brain of infected mice, and that the RNAemia in the sera and the viral load in the liver is rapidly increasing to reach very high viral RNA copies number at D3 pi. Furthermore, IN infection significantly delayed the appearance of virus in the liver because no viral RNA is detected until D3 pi, for which two mice ($10^{8-9}$ copies of vRNA/g) and a single serum were positive for RVFV ($\approx 10^8$ copies of vRNA/ml) (Fig 6). Notably, MRU25010-30 was detected in the brain of one mouse at D1, three mice at D2, and one mouse at D3 at levels comparable to those of SC infection ($\approx 10^{8-9}$ vRNA copies/g). These results confirmed that the brain is targeted early by MRU25010-30 regardless of the route of inoculation.

For strain MRU2687-3, we showed that two out three mice from SC-$10^3$ group were RVFV positive in the serum ($10^{9-10}$ vRNA copies/ml) and liver ($\approx 10^{11}$ vRNA copies/g) at D3 pi (Fig 6). Unfortunately, one mouse was found dead at D6 pi (and therefore not included in our analysis). We detected RVFV RNA only in one mouse liver ($\approx 10^9$ vRNA copies/g) when the two sera were negatives. By subcutaneous route, we did not find any mice with MRU2687-3 positive brain, whereas two mice were positive in the brain at D6 pi when infected intranasally ($\approx 10^{9-10}$ vRNA copies/ml). Furthermore, MRU2687-3 was detected at D3 pi (IN-$10^3$) in two out of three sera ($\approx 10^8$ vRNA copies/ml) and one out of three livers ($\approx 10^9$ vRNA copies/g), although at a lower level of viral RNA compared to that of SC-$10^3$ group. However, at D6 pi,

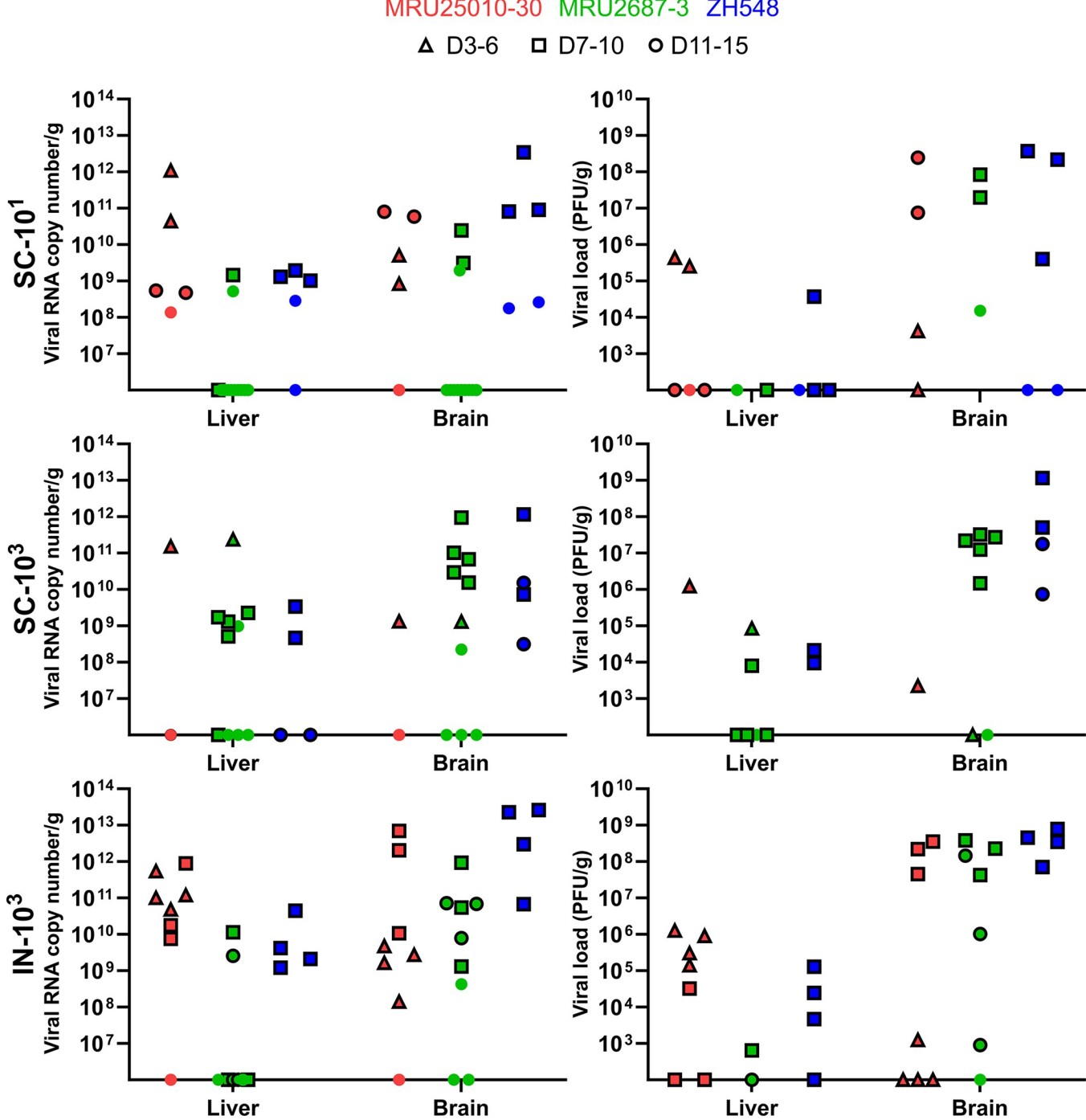

**Fig 5. Levels of viral RNA and titres in the liver and brain of BALB/c infected mice.** Liver and brain were collected from euthanised mice and levels of viral RNA measured. Values are expressed as viral RNA copy number per g of liver or brain (left). Positives samples for viral RNA were also titrated by plaque assays and viral titres are expressed as PFU/g (right). Triangles represent animal euthanised between D3 and D6 pi, squares between D7 and D10 pi, and circled dots between D11 and D15 pi. Surviving mice at D15 are represented with dots.

all mice infected with MRU2687-3 by IN route were positive (in both sera and livers) suggesting again that intranasal infection delayed the appearance of the virus in the liver even though no significant effect of the inoculation route on the virulence is observed for this strain (Fig 3).

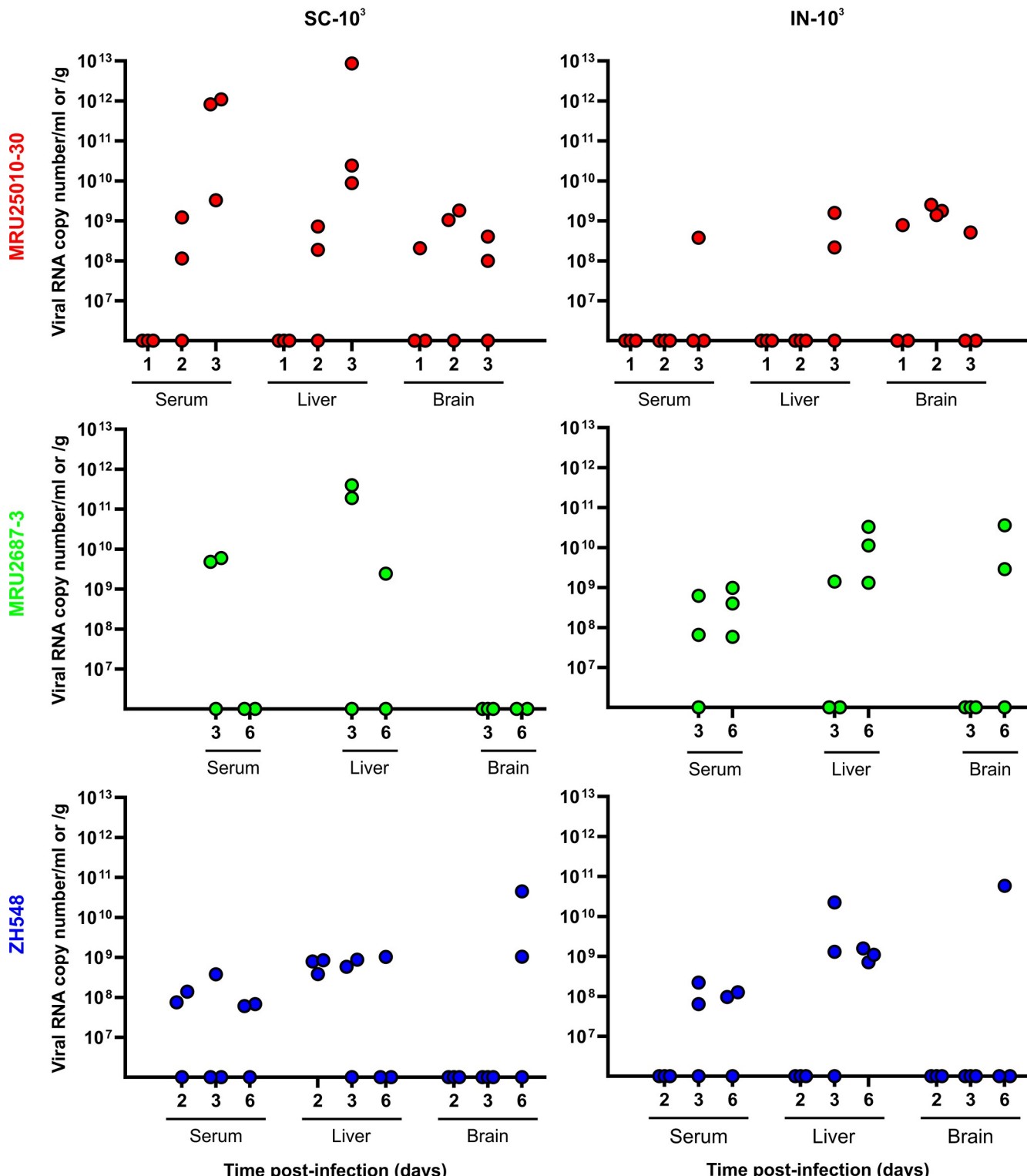

**Fig 6. Levels of RVFV RNA in the serum, liver, and brain of BALB/c mice at early stages of infection.** 6–8 weeks old female BALB/c mice were infected intranasally (IN) or subcutaneously (SC) with $10^3$PFU of MRU25010-30 (red), MRU2687-3 (green) or ZH548 (blue). Levels of viral RNA were measured by RT-qPCR targeting segment M and values are expressed as viral RNA copy number per ml (serum) or per g (liver and brain). Sera were collected at days 1, 2, and 3 (MRU25010-30, n = 3), days 3 and 6 (MRU2687-3, n = 3 but in SC-$10^3$ group where one mouse died at D6 pi), and days 2, 3, and 6 (ZH548, n = 3).

The same was generally observed in ZH548 infected mice, although its number of vRNA copies was substantially lower compared to both field strains in the livers and the sera at D3 pi for the SC-10$^3$ group (Fig 6). Altogether, these data confirmed that MRU2687-3 strain is able to replicate efficiently in the liver of infected mice and is detected in the brain at a later time point (> D6 pi) compared to MRU25010-30.

## Discussion

In this study, we investigated the biological properties of two RVFV field strains isolated in Mauritania to assess the impact of the viral genetic diversity on RVF replication capacity and pathogenesis. The first RVF outbreak in West Africa was recorded in 1987 and, since then, the virus has been particularly active in this region, causing multiple epidemics of varying magnitude and severity [24,42,51–53]. It is well-established that multiple RVFV introductions took place in West Africa and that several RVFV lineages are circulating in this area [54]. Phylogenetic analysis showed that MRU25010-30 belongs to lineage E containing Entebbe strain and a strain isolated in Zimbabwe (2973/74 strain), whereas MRU2687-3 is closely related to the lineage A consisting mainly of Egyptian and Namibian isolates including the ZH548 strain used as reference in this study [25]. Interestingly, Entebbe strain (E) inoculated intraperitoneally to CD-1 mice was more virulent than ZH501 strain (A) [26], and the lethal dose 50 of Entebbe was higher than of ZH501 in Wistar-Furth rats [55]. However, it is difficult to compare the data obtained in this study with those generated by others, as passage history (number of passages and/or cell lines used) of RVFV isolates may influence their virulence in mice. We found that BALB/c mice infected by MRU25010-30 strain succumbed earlier than the ones challenged with MRU2687-3, independently of the inoculation route. MRU25010-30 increased virulence is correlated with a higher RNAemia and viral load in the liver of the animals. Comparatively, MRU2687-3 and ZH548 pathogenesis is mainly linked with an active viral replication in the brain at later time point post-infection. These results are in line with disease features previously described in this mouse model, i.e. an acute-onset liver disease and a late-onset encephalitis [16]. Moreover, it has been previously shown that RVFV infectious dose is modulating the time of death [31,56,57]. We showed in this study that MRU25010-30 induced pathogenesis only slightly differed by using a lower dose, or *via* intranasal inoculation; the later delaying to some extent the infection of the liver.

Altogether, our data strongly suggest that MRU25010-30 has an increased potential to induce deadly liver damage compared to MRU2687-3 strain. These results are supported by the strong replication capacity of MRU25010-30 strain *in vitro* in HepaRG cells. Since, we also detected MRU25010-30 in the brain of infected mice as soon as 1 day pi, we cannot rule out that viral replication in this organ contributes to the pathogenesis induced by this strain. It is also known that RVFV targets the olfactory neurons lining the nasal tract, and BALB/c mice or Lewis rats exposed to aerosols or intranasally infected with RVFV have been shown to develop a much earlier and severe neuropathology than subcutaneously injected animals [17,58,59]. In our study, we did not observe difference in term of virulence between IN and SC conditions for MRU25010-30 and MRU2687-3 strains. Therefore, further studies are needed to investigate the dissemination and viral pathogenesis in the central nervous system depending on the route of inoculation and the field strain used.

By comparing protein sequences obtained in this study, we identified several amino acid residues substitutions of interest that differ between MRU25010-30 and MRU2687-3/ZH548 stains, in particular within NSm (R42G), Gn (K384T), and L (D157G; D407G; G411S; T2033A) and could contribute to the differences observed *in vivo* and *in vitro*. Notably, amino acid residues D157 and D407 within the L protein are common between MRU25010-30 and

Entebbe strains. L protein is the viral RdRp involved in the transcription and replication processes as well as cap-snatching mechanism [60,61]. D157G is located in the endonuclease domain while D407G and G411S are part of the Arch domain, within PA-C like domain [61]. Finally, T2033A is located at the C terminal part of the RdRp within the PB-2 like domain [61]. The role of NSm is still poorly understood but it has been shown to be dispensable *in vitro* for viral replication and a mutant virus unable to express NSm/NSm' is greatly attenuated in C57BL/6 mice [31,32,62,63]. Interestingly, it was shown that RVFV NSm proteins localize in the outer mitochondrial membrane of infected mammalian cells and display anti-apoptotic activity due to a domain located in the C terminal part of the protein [29,30]. However, R42G substitution is located in the N terminal region of NSm and, therefore, it should not impact its anti-apoptotic function. Gn protein, together with Gc, is involved in the viral entry processes as well as viral particles morphogenesis and egress [64–69]. K384T is located in the domain B at the surface of Gn and could therefore interfere with Gn functions [70,71]. Both mutations (R42G and K384T) are also part of p78 that has been shown to be essential for RVFV dissemination in *Aedes aegypti* mosquitoes [31,72]. The lack of p78 expression does not modulate RVFV virulence in mice; however, a recent study showed that the quantity of p78 produced by RVFV impacts virus replication rates in human macrophages as well as its virulence in infected mice [31,34]. Notably, we did not identify amino acid residue substitutions drastically changing the charge of the R-group in NSs protein, the main RVFV virulence factor [37,73–75]. V262A mutation is located at the X position of the ΩXaV motif, known to be required for NSs nuclear filament formation and function [36]. Although this amino acid may be involved in the difference observed between the two strains, our results suggest that the increased growth properties of MRU25010-30 in A549 cells is not solely linked to IFN pathway, known to be inhibited by NSs [37,38].

Finally, it has been shown that the genetic diversity of viral populations may affect disease outcome or virulence of several viruses such as hepatitis C virus, poliovirus or WNV [76–78], and that RVFV subpopulations are modulating its virulence in CD-1 infected mice [26,79]. In this study, we characterized RVFV strains minimally passaged after their isolation and without any plaque purification step to keep their intra-strain genetic diversity. We identified viral subpopulations in the three viral strains studied. The vast majority of the intra-strain diversity was found in Gn protein sequences. Interestingly, deep-sequencing analysis combined with RACE-PCR approach allowed us to unveil a subpopulation at nucleotide position 10, which can be either an uracil or a cytosine, in segment M (antigenome) of both MRU2687-3 and ZH548 strains. The uracil 10 leads to an additional initiation AUG codon located upstream of AUG1, which has already been described for MP-12 vaccine strain [80].

To date, given the number of RVFV strains tested in this study and the fact that no viral determinant can yet be associated clearly with the virulence observed in mice, it is difficult to extrapolate our data to other RVFV strains or lineages. Indeed, the impact of the intra- and inter-strain genetic diversity on the striking differences observed between the two field strains remains unknown. However, it will be now possible to develop specific reverse genetics systems, based on our sequencing data, to decipher the effect of quasi-species as well as amino acid substitutions on the RVFV-induced pathogenesis, replication, and interaction with host immune response, such as IFN signalling pathways. Along this line, our comparative study highlights the importance of strengthening such approaches to characterize further RVFV molecular determinants associated with its virulence in mammalian hosts or vector transmission. It implies collecting more viral isolates and generating full-genome sequencing when possible. This effort would greatly help implement control strategies adapted to the threat posed by emerging RVFV strains in a given area.

## Supporting information

**S1 Data. Numerical data used to generate Figs 2, 3, 4, 5, 6, S5, and S6.**
(XLSX)

**S1 Table. Mapping statistics of the high throughput sequencing data.**
(PDF)

**S2 Table. Amino acid substitutions observed between the consensus sequences of MRU25010-30, MRU2687-3, and ZH548 strains.** Amino acid substitutions are classified by viral segment and related protein. Their position was determined relative to the known start codon. Note that the numbering of M segment proteins (NSm, Gn, and Gc) starts from AUG1 used to translate p78. Amino acid residues can be classified in four groups based on their polarity (non-polar, polar with no charge on R group, polar with negative charge on R group, and polar with positive charge on R group). Amino acid residues conserved between MRU25010-30 and MRU2687-3 are uncoloured. Non-conserved amino acid residues between these two strains but identical between MRU25010-30 and ZH548 strains are coloured in light grey. Non-conserved amino acid residues between MRU25010-30 and MRU2687-3, and from the same group are coloured in grey. Non-conserved amino acid residues and from a different group are coloured in dark grey.
(PDF)

**S1 Fig. Coverage and Shannon entropy plots for L segments.** Coverage (1) and Shannon entropy (2) for L segments of MRU2687-3 (A), MRU25010-30 (B), and ZH548 (C).
(PDF)

**S2 Fig. Coverage and Shannon entropy plots for M segments.** Coverage (1) and Shannon entropy (2) for M segments of MRU2687-3 (A), MRU25010-30 (B), and ZH548 (C).
(PDF)

**S3 Fig. Coverage and Shannon entropy plots for S segments.** Coverage (1) and Shannon entropy (2) for S segments of MRU2687-3 (A), MRU25010-30 (B), and ZH548 (C).
(PDF)

**S4 Fig. DNA sequencing chromatograms of RACE-PCR products amplified from 5'UTR of anti-genomic M segments.** Nucleotides at position 10 is either an uracile (U) or a cytosine (C) in MRU2687-3 and ZH548 strains.
(PDF)

**S5 Fig. Effect of RVFV infection on BALB/c mice body weight.** Mice have been weighted before infection and then every day during the course of the experiment. Weight percentage was calculated relative to that at Day 0. Black dots represent the last measure recorded before the death of the animal.
(PDF)

**S6 Fig. Kinetics of seroconversion of mice infected with MRU25010-30, MRU2687-3, and ZH548 strains.** Anti-RVFV antibodies within mice sera were detected using in-house IgM and IgG ELISAs (Enzyme-Linked ImmunoSorbent Assay), as previously described [41,48]. Briefly, RVFV antigens were prepared from RVFV MP12 strain infected VeroE6 cells (MOI = 0,01; 2 days post-infection). For IgM detection, 96 -well plates (Nunc Maxisorp, Thermo Fisher Scientific) were coated with rabbit anti-mouse IgM antibody (100 μL/well, 1:400 dilution; Sigma, SAB3701197) and incubated with 100 μL/well of 1:100 dilution of mice sera. RVFV antigens were subsequently detected with hyperimmunised sera from hamster

infected with ZH501 strain and Goat anti-Hamster IgG (H+L)-HRP (Horseradish Peroxidase) conjugated antibody. For IgG detection, plates were coated with RVFV antigens, further incubated with 100 µL/well of 1:100 dilution of mice sera and subsequently with HRP-conjugated rabbit anti-mouse IgG (whole molecule, 1:5000, Sigma, A9044). HRP enzymatic activity was revealed using TMB substrate (Thermo Fischer Scientific). Optical density at 450 nm (OD450) was measured using a TECAN microplate reader. ELISA measurement of IgM and IgG antibodies of mice infected with field strains or ZH548. The sera were collected at days 3, 4–6, 7–10, and 11–15 pi. At the indicated day, black bars represent dead mice, grey bars seroconverted mice, and white bars mice with non-detectable IgM or IgG antibodies.
(PDF)

## Acknowledgments

We acknowledge the contribution of the BSL3 platform of SFR BioSciences Gerland Lyon Sud (UMS3444/US8). The authors would like to thank Dr Kyriaki Nomikou, Dr Geneviève Conejero and Johan Deniaud for their implication in this work.

## Author Contributions

**Conceptualization:** Philippe Marianneau, Catherine Cêtre-Sossah, Frédérick Arnaud, Maxime Ratinier.

**Data curation:** Mehdi Chabert, Sandra Lacôte, Philippe Marianneau, Aurélie Pédarrieu, Ahmed Bezeid El Mamy Beyatt, Vattipally B. Sreenu, Catherine Cêtre-Sossah, Frédérick Arnaud, Maxime Ratinier.

**Formal analysis:** Mehdi Chabert, Sandra Lacôte, Philippe Marianneau, Aurélie Pédarrieu, Ahmed Bezeid El Mamy Beyatt, Vattipally B. Sreenu, Catherine Cêtre-Sossah, Frédérick Arnaud, Maxime Ratinier.

**Funding acquisition:** Catherine Cêtre-Sossah, Frédérick Arnaud, Maxime Ratinier.

**Investigation:** Philippe Marianneau, Catherine Cêtre-Sossah, Frédérick Arnaud, Maxime Ratinier.

**Methodology:** Mehdi Chabert, Sandra Lacôte, Marie-Pierre Confort, Noémie Aurine, Jenna Nichols, Vattipally B. Sreenu, Ana da Silva Filipe, Marie-Anne Colle, Bertrand Pain, Frédérick Arnaud, Maxime Ratinier.

**Project administration:** Maxime Ratinier.

**Resources:** Aurélie Pédarrieu, Baba Doumbia, Mohamed Ould Baba Ould Gueya, Habiboullah Habiboullah, Ahmed Bezeid El Mamy Beyatt, Modou Moustapha Lo.

**Supervision:** Philippe Marianneau, Marie-Pierre Confort, Ana da Silva Filipe, Bertrand Pain, Catherine Cêtre-Sossah, Frédérick Arnaud, Maxime Ratinier.

**Validation:** Philippe Marianneau, Ana da Silva Filipe, Bertrand Pain, Catherine Cêtre-Sossah, Frédérick Arnaud, Maxime Ratinier.

**Writing – original draft:** Mehdi Chabert, Frédérick Arnaud, Maxime Ratinier.

**Writing – review & editing:** Mehdi Chabert, Sandra Lacôte, Philippe Marianneau, Marie-Pierre Confort, Ahmed Bezeid El Mamy Beyatt, Modou Moustapha Lo, Jenna Nichols, Vattipally B. Sreenu, Ana da Silva Filipe, Bertrand Pain, Catherine Cêtre-Sossah, Frédérick Arnaud, Maxime Ratinier.

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
