## [Decision Letter · Decision Letter 0]

18 Jun 2024

Dear Dr Ratinier,

Thank you very much for submitting your manuscript "Comparative study of two Rift Valley fever virus field strains circulating in Mauritania in 2010 and 2013 reveals the high virulence of the MRU25010-30 strain isolated from camel" for consideration at PLOS Neglected Tropical Diseases. As with all papers reviewed by the journal, your manuscript was reviewed by members of the editorial board and by several independent reviewers. In light of the reviews (below this email), we would like to invite the resubmission of a significantly-revised version that takes into account the reviewers' comments. 

Thank you for your submission. Your manuscript draft has been reviewed by three peer reviewers. Reviewer two in particular has provided a number of helpful recommendations and suggestions. You are encouraged to review all reviewer comments and make appropriate edits for a secondary review. I agree with the reviewers comments generally. I feel the manuscript can be improved with additional editing for grammar and parsimony. The authors are encouraged to consider additional limitations to their study given the number of isolates characterized in this study.

We cannot make any decision about publication until we have seen the revised manuscript and your response to the reviewers' comments. Your revised manuscript is also likely to be sent to reviewers for further evaluation.

Sincerely,

Brian Byrd

Guest Editor

David Safronetz

Section Editor

Thank you for your submission. Your manuscript draft has been reviewed by three peer reviewers. Reviewer two in particular has provided a number of helpful recommendations and suggestions. You are encouraged to review all reviewer comments and make appropriate edits for a secondary review. I agree with the reviewers comments generally. I feel the manuscript can be improved with additional editing for grammar and parsimony. The authors are encouraged to consider additional limitations to their study given the number of isolates characterized in this study.

Reviewer's Responses to Questions

**Key Review Criteria Required for Acceptance?**

**Methods**

-Are the objectives of the study clearly articulated with a clear testable hypothesis stated?

-Is the study design appropriate to address the stated objectives?

-Is the population clearly described and appropriate for the hypothesis being tested?

-Is the sample size sufficient to ensure adequate power to address the hypothesis being tested?

-Were correct statistical analysis used to support conclusions?

-Are there concerns about ethical or regulatory requirements being met?

Reviewer #1: These have been described in sufficient detail, making it easy to follow the results obtained, which are in turn well presented, both in figures and words.

There are a few grammatical inaccuracies, resulting from idiosyncrasies of the English language, but these can be easily rectified at the editing stage.

Reviewer #2: Please see my comments in Summary and general comments section. This study needs to confirm the presence of replicating RVFV is some of the samples.

Reviewer #3: -Are the objectives of the study clearly articulated with a clear testable hypothesis stated? Yes

-Is the study design appropriate to address the stated objectives? Yes

-Is the population clearly described and appropriate for the hypothesis being tested? yes

-Is the sample size sufficient to ensure adequate power to address the hypothesis being tested? No

-Were correct statistical analysis used to support conclusions? yes

-Are there concerns about ethical or regulatory requirements being met? no

**Results**

-Does the analysis presented match the analysis plan?

-Are the results clearly and completely presented?

-Are the figures (Tables, Images) of sufficient quality for clarity?

Reviewer #1: These are well presented and are confined the data generated in the work. 

The investigations were done both in vivo and in vitro, both producing complementary results.

Reviewer #2: Yes, but please see my comments in Summary and general comments section.

Reviewer #3: -Does the analysis presented match the analysis plan? yes

-Are the results clearly and completely presented? yes

-Are the figures (Tables, Images) of sufficient quality for clarity? yes

**Conclusions**

-Are the conclusions supported by the data presented?

-Are the limitations of analysis clearly described?

-Do the authors discuss how these data can be helpful to advance our understanding of the topic under study?

-Is public health relevance addressed?

Reviewer #1: These are supported by the data obtained

Reviewer #2: Please see my comments in Summary and general comments section.

Reviewer #3: -Are the conclusions supported by the data presented? no, it is difficult to conclude high virulence with only a single sample

-Are the limitations of analysis clearly described? no

-Do the authors discuss how these data can be helpful to advance our understanding of the topic under study? yes

-Is public health relevance addressed? yes

**Editorial and Data Presentation Modifications?**

Reviewer #1: The manuscript is well-written; however, it could be improved to make it clearer and more accurate. For example, the use of the words virulence (lines 2 and 53) and pathogenicity (line 79 & so on) need to be context sensitive, such that the aspects relating to the virus and those relating to the host can be clearly understood. 

The authors should bear in mind also that recent field isolates of pathogens tend to be more pathogenic. This decreases with passages and/or propagation in vitro or in vivo. Although the factors which underlie this are not understood, some could confound observations such as the ones made in this study.

Reviewer #2: It is difficult for people for whom English is a second language to write a quality English manuscript. Please see my comments in Summary and general comments section.

Reviewer #3: The mutation analysis should be enhanced to see if correlations with virulence can be made.

**Summary and General Comments**

Reviewer #1: Specific 

Line

Comment

34-35

isolated from, not in

54 and 124:

it is the genomes of the viruses, not the viruses, that were sequenced 

83

outbreak refers to disease, in this case Rift Valley fever, not RVFV

95

change “is” to “has been”.

124

insert the genomes of between the words sequenced and these

321

acids should be acid

433

replace have with had

435 & 472

replace has been with was

523

replace participates with contributes, etc

Reviewer #2: General Comment: This study evaluated the relative virulence of two strains of RVFV isolated in Mauritania in cell culture and mice. It also did deep sequencing to see if viral RNA mutations might be connected with virulence. I know that it is difficult for authors for whom English is not their primary language to write in English, and I assume that some of the errors commented on below were based on the choice of the wrong English word. One quick general comment, throughout the manuscript, “have been…” should be replaced with “were…” More importantly, as indicated in the comments below, detecting viral RNA does not mean that any infectious virus was present. Therefore, determining a viremia via RT-qPCR will often give misleading information and greatly extend the actual viremic period. I did not see anywhere in the manuscript that infectious virus was detected from any of the samples. Unfortunately, I have seen several studies where pieces of viral RNA were detected and the authors claimed to have found virus, but upon checking, no infectious virus was present. Some of these sample need to be tested for infectious RVFV.

Specific Comments: 

1. Title: Minor, but shouldn’t it be “isolated from a camel?”

2. Lines 73-80: Shouldn’t you also mention that humans can be infected with RVFV by the bite of an infectious mosquito?

3. Line 90: Yes, there were 63 recognized and reported human cases, but how many additional human infections/cases were there that were not recognized or reported. As stated in the current paper, it implies that the case fatality rate was 21%, which is very high for RVF in humans. The cited paper also stated, “a total of 63 cases among humans, including 13 deaths, had been officially reported, but the true number is probably much higher due to the remoteness of the affected area.” I think you should include a statement that the number of cases was likely to be underreported and was actually much greater. 

4. Lines 229-233: What was the volume of inoculum used to infect mice either IN or SC? If you are going to repeat the methods on lines 376-378. You should also include the volumes. Did you use the same volume for SC as for IN? If so, that was a lot to inject IN in a mouse and may have done some tissue damage resulting in some SC inoculation. If not, then did you use difference concentrations of stock virus?

5. Lines 379-384 (Figure 3): several comments:

 a. Why write “10PFU (10) or 103PFU (103) of…” instead of “10^1 or 10^3 PFU of…?”Please note that I had to use the “^” character as I cannot use a superscript in the review.

 b. “eight weeks old…” should be “8 weeks old…”

 c. I assume that there were 12 mice for each dose/route for the two Mauritanian strains of RVFV. Is that correct?

 d. I was a little confused by how the significant differences were indicated. Looking at SC-10 (note, that really should be 10^1 to be consistent with the way you are indicating titer), it appears that the survival of mice inoculated SC with MRU2687-3 was not significantly different than those inoculated with ZH548 as indicated by brackets. However, the two survival rates are clearly significantly different.

 e. Here and throughout the manuscript, “p<0,05” should be “p<0.05” as in English usage, a period, rather than a comma, is used to indicate a decimal. Please change all.

6. Lines 392-394: Yes, the LD/50 was 9 days for both doses. However, the mortality rate was 67% for those inoculated with10^1 PFU as compared to 100% for those inoculated with 10^3 PFU, so there does appear to be a dose effect on mortality.

7. Lines 394-395: For both routes the mortality rate was 100% and the LD/50’s were 8 versus 9 days, i.e., one mouse died one day earlier in one group. Is that really significant?

8. Line 404: Yes, you used RT-PCR to detect RVFV RNA, but were any of the samples actually checked to confirm that they contained infectious RVFV. Unfortunately, noninfectious pieces of RNA may persist in the blood producing a false positive if only RNA is sampled. I would be surprised if any of the mice were viremic on day 5 after infection despite the presence of RVFV RNA.

9. Lines 415-416: Again, checking for RNA does not check for virus or viremia. Did any of the mice contain virus in their blood when they died on or after day 5?

10. Lines 417-423 (Figure 4): Again, why write “10PFU (10) or 103PFU (103) of…” instead of “10^1 or 10^3 PFU of…?”

11. Line 427: Did any of these six mice that failed to produce IgM have a viremia, i.e., were they infected with RVFV?

12. Lines 434-437: Why weren’t you able to test more mice? Based on Table 3, six mice infected with 10^3 PFU of MRU25010-30 SC died on day 4? Even more importantly, did you confirm that the RNA detected in that mouse actually represented infectious RVFV?

13. Lines 452-359: If the purpose of this study was to compare the two strains of RVFV, why did you test viremias and the presence of virus in livers and brains on different days for the two strains? Again, the mere presence of viral RNA does not mean that any infectious virus was present.

14. Lines 515-516: Despite what was said here, time to death was also associated with dose and mice given the larger dose died sooner.

15. Lines 521-530: Interesting. Looking at Figure 6, how could there be virus in the brain of one of the SC inoculated mice on day 1 if there was no virus detected in the blood of that mouse? I would bet that there was an error in the laboratory.

16. Lines 566-569: Was this a subpopulation within the MRU2687-3 strain that differed or was it the 5’UTR of segment M at position 10 of the MRU2687-3 strain that differed?

17. References: These need to be formatted properly.

 a. Only the first word and proper nouns in a reference title should be capitalized. See references 2, 13, 25 and many others. 

 b. Proper nouns need to be capitalized. See references 5, 8, 9, and many others.

 c. For scientific names, the genus should be capitalized, the species not capitalized, and the species name should be in italics. See references 3 and 39. See also reference 20 where there are extra spaces before and after the scientific name.

 d. Why is the name of the editor of the various PLoS journals included

 e. Shouldn’t PLOS ONE or PLoS ONE be PLoS One? See references 52, 54, and 79?

Minor Comments:

18. Line 34: “has been isolated in camel” should be “was isolated from a camel…” Similarly, “isolated in goat” should be “isolated from a goat…”

19. Lines 38, 39: All results presented in a paper should be in the past tense, so “analysis shows…” should be “analysis showed…” and “we show that…” should be “we showed that…” please check the remainder of the manuscript as there are numerous results and conclusions in the present tense.

20. Line 84: Shouldn’t “an RVFV outbreak…” be “a RVFV outbreak…”

21. Line 116: What are “ORFs?” I know that you are referring to open reading frames, but if you are gong to establish “nts” then you should establish ORF for “open reading frames.” As it is used again.

22. Line 148: The “2” in “CO2” should be a subscript.

23. Line 150: As the strain was only isolated one time, it should be “…strain was isolated…” Same comment for lines 152, 154, and 155.

24. Line 180: It is probably better to use “were extracted…” rather than “have been extracted… Again, throughout the manuscript, virtually all of the ”have been” should be “were…”

25. Line 199: why use “75nt” instead of “75nts? Earlier, line 100, you established nts to mean nucleotides.

26. Line 215: Because “12” is being used to modify “well,” it should be “12-well format…” or probably better to simply state, “12-well pates and…”

27. Line 218: Why capitalize “Plaque Forming Units?” It should be “plaque-forming units (PFU,…” Note the hyphen between plaque and forming.

28. Line 222: Again, why capitalize “Tissue Culture Infectious Dose?” These are not proper nouns. Also, the 50 in TCID50 should be a subscript.

29. Line 231: As there was more than one dose, it should be “were either…”

30. Line 232: Numbers, not of measurement, less than 10 should be written out, so this should be “two or three animals…” and six animals…” See also line 244, 252, 366, 457, and many others.

31. Line 240: Despite the above comment, this should be 6 to 8 weeks as weeks are a unit of measurement. Please also see line 229 and others.

32. Line 270: Again, it is probably better to use “were obtained…”

33. Line 282: Should “99%” be “99.0%” as the others are given to a tenth of a percent?

34. Line 335: Why establish HepaRG here? It should have been established the first time it was use on line 137.

35. Line 349: As was used later in this same sentence, use “were infected” instead of “have been infected…”

36. Line 357: why establish “SD” if it is not used again?

37. Line 365: results should always be in the past tense, so this should be, “we showed that…” See also line 368 where “reaches…” should be “reached…”, line 470 where “we show…” should have been “we showed…” and several others. Please check the entire manuscript.

38. Line 377: Why establish IN and SC again here? They were established on line 230.

39. Line 381: Why is it “Six-eight weeks” here and “6-8 weeks…” in Figures 4 and 6?

40. Line 410: “figure 3” should be “Figure 3.”

41. Lines 435, 460, and many others: It is more accurate to use, “was…” rather than “has been…” 

42. Line 470: Why is there a space after the “7” in MRU2687 -3?

43. Line 536: Why reestablish “RdRp” when it was already established on line 100?

44. Line 509: Because there was more than one mouse, “than the one…” should be “than the ones…”

Reviewer #3: The experiment was conducted well and the results clearly presented. However because of only one sample being analysed, the conclusion that it is highly virulence needs additional support, for example from mutation analysis.

PLOS authors have the option to publish the peer review history of their article (what does this mean?). If published, this will include your full peer review and any attached files.

Reviewer #1: Yes: Prof. Phelix A.O. Majiwa

Reviewer #2: No

Reviewer #3: No
---

## [Decision Letter · Decision Letter 1]

31 Oct 2024

PNTD-D-24-00500R1Comparative study of two Rift Valley fever virus field strains originating from MauritaniaPLOS Neglected Tropical Diseases Dear Dr. Ratinier, Thank you for submitting your manuscript to PLOS Neglected Tropical Diseases. After careful consideration, we feel that it has merit but does not fully meet PLOS Neglected Tropical Diseases's publication criteria as it currently stands. Therefore, we invite you to submit a revised version of the manuscript that addresses the points raised during the review process. Please submit your revised manuscript within 30 days Nov 30 2024 11:59PM. If you will need more time than this to complete your revisions, please reply to this message or contact the journal office at plosntds@plos.org. Please include the following items when submitting your revised manuscript:*
A rebuttal letter that responds to each point raised by the editor and reviewer(s). You should upload this letter as a separate file labeled 'Response to Reviewers'. This file does not need to include responses to any formatting updates and technical items listed in the 'Journal Requirements' section below.*
A marked-up copy of your manuscript that highlights changes made to the original version. You should upload this as a separate file labeled 'Revised Manuscript with Track Changes'.*
An unmarked version of your revised paper without tracked changes. You should upload this as a separate file labeled 'Manuscript'. If you would like to make changes to your financial disclosure, competing interests statement, or data availability statement, please make these updates within the submission form at the time of resubmission. Guidelines for resubmitting your figure files are available below the reviewer comments at the end of this letter. We look forward to receiving your revised manuscript. Kind regards, Brian ByrdGuest EditorPLOS Neglected Tropical Diseases David SafronetzSection EditorPLOS Neglected Tropical Diseases

Shaden Kamhawi

co-Editor-in-Chief

Paul Brindley

co-Editor-in-Chief

 **Journal Requirements:** **Additional Editor Comments (if provided):** Thank you for your improved manuscript. The reviewers are also in agreement that this revision is much improved. Reviewer 2 has pointed out some additional areas where minor revisions are required and provided helpful edits. Please respond to each of Review 2's suggestions and edits. Please note that some comments may not require edits to the manuscript, but if edits would provide additional clarity and precision, you should strongly consider revisions.**Reviewers' comments:** Reviewer's Responses to Questions

**Key Review Criteria Required for Acceptance?**

**Methods**

-Are the objectives of the study clearly articulated with a clear testable hypothesis stated?

-Is the study design appropriate to address the stated objectives?

-Is the population clearly described and appropriate for the hypothesis being tested?

-Is the sample size sufficient to ensure adequate power to address the hypothesis being tested?

-Were correct statistical analysis used to support conclusions?

-Are there concerns about ethical or regulatory requirements being met?

Reviewer #1: These have been described in sufficient detail. The suggestions made in the first version have been satisfactorily addressed. In particular, the grammatical inaccuracies, have been removed

Reviewer #2: Please see general comments

Reviewer #3: All have been met satisfactorily.

**Results**

-Does the analysis presented match the analysis plan?

-Are the results clearly and completely presented?

-Are the figures (Tables, Images) of sufficient quality for clarity?

Reviewer #1: These are well presented and are confined the data generated in the work. Suggestions made by this and other reviewers have been adequately addressed and/or incorporated.

Reviewer #2: Please see general comments

Reviewer #3: All met satisfactorily.

**Conclusions**

-Are the conclusions supported by the data presented?

-Are the limitations of analysis clearly described?

-Do the authors discuss how these data can be helpful to advance our understanding of the topic under study?

-Is public health relevance addressed?

Reviewer #1: These are supported by the data obtained.

Reviewer #2: Please see general comments

Reviewer #3: Yes

**Editorial and Data Presentation Modifications?**

Reviewer #1: This reviewer has nothing useful to add

Reviewer #2: Please see general comments

Reviewer #3: Accept

**Summary and General Comments**

Reviewer #1: This manuscript contains information that is of interest to the readership of this journal. The revised version is acceptable for publication.

Reviewer #2: General Comment: The revised manuscript is much improved. Thanks for including the data on the presence of actual infectious virus. Below, in addition to a few comments that need to be addressed, are some suggested word changes to improve the “English” in the manuscript.

Specific Comments:

1. Lines 443-444: Given that five of the six mice that failed to produce IgM also failed to produce detectable RNA in their sera, isn’t it very likely that those mice were never infected with the MRU2687-3 strain of RVFV despite being inoculated with that virus? Did you detect virus or RNA from any of the organs? If not, I would assume that it is more likely that those mice were never infected with the MRU2687-3 strain of RVFV, even if they had been inoculated with that strain.

2. Line 457: Were you testing for MRU25010-30 virus or MRU25010-30 RNA. I would indicate that it was RNA.

3. Line 509: Unfortunately, Figure 6 is a bit misleading. Looking at the figure, it appears that MRU25010-30 replicates faster in the serum and liver than in mice inoculated SC with the MRU2687-3 strain. However, this difference is due to no mice inoculated with MRU2687-3 being tested before day 3. If the goal of that study was to compare the two strains, they should have been tested at the same time periods.

4. Line 529: What are “WF rats?”

5. Lines 351-353: If the mouse inoculated IN with MRU25010-30 had no virus in the serum on day 1 and yet had a high brain titer, how did the virus get to the CNS if not directly from the olfactory tract?

Minor comments:

6. Line 36: Why stablish “RACE-PCR” if it is not used again in the abstract? You should not establish an abbreviation if it is not used again. See also line 154 for “ISRA-LNERV” and line 155 for “CIRAD.”

7. Line 53-54: When writing a list, i.e., “comment a, comment b, and comment c” always include a “,” after the item just before the “and” so “replicative capacities and transmission…” should be “replicative capacities, and transmission…” See also line 69, 71, 88, 107, 123, 135, 141, 167, 275, 291, 304, and others.

8. Line 91: Shouldn’t “under evaluated” be “under reported?”

9. Line 162: “Two or three-days…” should be “Two or 3-days…” as numerals are used for all units of measurement except at the beginning of a sentence. Note, on line 244, you wrote “6 to 8 weeks old…” at the beginning of the sentence. You need to be consistent. I would have used “Six- to 8-week-old female…”

10. Line 246: I thought that you had corrected this. It should be “either 10^1 or 10^3 PFU.” See also lines 400 and 436 where “10PFU (10^1) or 10^3PFU (10^3)”…” should be “10^1 or 10^3 PFU…”

11. Line 258: “1000 PFU…” should be “10^3 PFU…” Remember, I cannot use a superscript in this program.

12. Line 273: “3 times…” should be “three times…”

13. Line 364: Is it “enhanced viral ability to replicate” or “enhanced ability to replicate in viral cells?”

14. Line 384: “fifteen times more…” should be “15 times more…”

15. Line 416: It should be “with MRU2687-3 or ZH548,…” because you did not inoculate and mice with both MRU2687-3 and ZH548.

16. Line 444: Shouldn’t “MRU2687…” be “MRU2687-3…?”

17. Line 454: Should “≈10^10-10^12 versus ≈10^8-10^9 copies…” be “≈10^10-12 versus 10^8-9 copies…?” See also line 479.

18. Lines 464-465: It might be better to state, “In contrast, the pathogenicity…”

19. Line 478: “two out three mice…” should be “two out of three mice…”

20. Line 507: Yes, it may enter “into” the brain later, but it is detected “in the brain…”

21. Line 526: Is “regrouping” the right word? Would it be better to write, “lineage A, consisting mainly of Egyptian and Namibian isolates, including the ZH548 strain…?”

22. Line 529: Why establish “LD50” if you do not use it again?

23. Line 785: Only the first letter of the first word (and proper nouns) should be capitalized. See also line 803 for Bovine Viral diarrhea virus, where neither bovine nor viral is a proper noun.

24. Line 899: “Rift Valley” should be capitalized.

Reviewer #3: The authors have revised the manuscript well and addressed most of the concerns. The only part that has not been fully addressed is why the lineage assignments could not be replicated with other tools. This however doesn't prevent the current publication but require a followup by the RVF experts.

PLOS authors have the option to publish the peer review history of their article (what does this mean?). If published, this will include your full peer review and any attached files.

Reviewer #1: **Yes: **Phelix A.O. Majiwa

Reviewer #2: No

Reviewer #3: No

---

## [Editor Report · Decision Letter 2]

26 Nov 2024

Dear Dr Ratinier,

We are pleased to inform you that your manuscript 'Comparative study of two Rift Valley fever virus field strains originating from Mauritania' has been provisionally accepted for publication in PLOS Neglected Tropical Diseases.

Best regards,

Brian Byrd

Guest Editor

David Safronetz

Section Editor

Shaden Kamhawi

co-Editor-in-Chief

Paul Brindley

co-Editor-in-Chief

---

## [Editor Report · Acceptance letter]

2 Dec 2024

Dear Dr Ratinier,

We are delighted to inform you that your manuscript, "Comparative study of two Rift Valley fever virus field strains originating from Mauritania," has been formally accepted for publication in PLOS Neglected Tropical Diseases.

Best regards,

Shaden Kamhawi

co-Editor-in-Chief

Paul Brindley

co-Editor-in-Chief
